# On-Chip Multichannel Dispersion Compensation and Wavelength Division MUX/DeMUX Using Chirped-Multimode-Grating-Assisted Counter-Directional Coupler

**Zhixiao Lv, Jiangbing Du *** and **Zuyuan He**

State Key Laboratory of Advanced Optical Communication Systems and Networks, Shanghai Jiao Tong University, Shanghai 200240, China; zuyuanhe@sjtu.edu.cn (Z.H.)
*   Correspondence: dujiangbing@sjtu.edu.cn

**Abstract:** On-chip optical dispersion compensation and wavelength division multiplexing/demultiplexing (WDM) are highly demanded functions for optical communications. In this work, we proposed a multichannel dispersion compensation structure based on chirped multimode grating within a counter-directional coupler (CMG-CDC). Simultaneous wavelength division multiplexing and de-multiplexing can be realized within a compact footprint. A device design for four-channel CMG-CDC at the C/L (1530–1565 nm) band is presented with a channel spacing of 20 nm assisted by a grooved multimode waveguide structure. The average dispersion for all channels is about −2.25 ps/nm with a channel bandwidth of about 3.1 nm. The device is highly compact and highly scalable, which makes it rather convenient for increasing the group velocity dispersion (GVD) and channel number, indicating flexible applications for versatile systems, including typically coarse wavelength division multiplexer four-lane (CWDM4) transceivers.

**Keywords:** optical interconnection; photonic integration; dispersion compensation

## 1. Introduction

In recent years, the continuously increasing amount of data generated by applications like data center, cloud computing, and so on has been significantly boosting the demand of high-speed optical communication for short reach, like optical interconnects, and long reach, like optical fiber transmissions. The increased baud rate directly leads to higher sensitivity for chromatic dispersion, which is one of the main factors limiting optical transmission distance and capacity due to the inter-symbol interference caused by pulse broadening. Therefore, versatile methods have been presented for compensating chromatic dispersion in high-speed optical communications, such as chirped fiber Bragg gratings (CFBGs) [1], dispersion compensating fibers (DCFs) [2], digital signal processing (DSP), and so on [3]. Among them, on-chip optical dispersion compensation for multichannel operation is highly demanded due to advantages like small footprint, low cost, and high-volume fabrication compatibility. Particularly, on-chip chirped Bragg gratings have been widely investigated as an alternative of CFBGs [4]. However, this grating works on reflection mode, so an optical circulator is required to separate the reflected signal, which is difficult for on-chip integration. Many solutions have been proposed to try to overcome this challenge, such as 3 dB Y-branch [5], adiabatic directional coupler (ADC) [6], and grating-assisted counter-directional couplers [7,8].

However, for chirped Bragg grating, the maximal group delay is limited by the grating length. Thus, a large amount of dispersion and wide working bandwidth for multiple channels usually mean a long device length and a large footprint. Several compact solutions such as Moire structure [9], metamaterial structure [10], and three-waveguide structure [11] have been demonstrated to multiply the channel numbers without increasing the coupling lengths, but only for uniform Bragg gratings. Therefore, a chirped Bragg grating supporting

multichannel dispersion compensation with the capability of simultaneous wavelength division multiplexing/demultiplexing (WDM) would be highly useful, which is exactly the main objective of this study.

In this work, a method for on-chip multichannel dispersion compensation with simultaneous wavelength division multiplexing and demultiplexing is presented and investigated. The proposed dispersion compensator is based on chirped multimode grating within a counter-directional coupler (CMG-CDC). A device design for four-channel CMG-CDC at the C/L band (1530–1565 nm) is presented with a channel spacing of 20 nm assisted by a grooved multimode waveguide structure. The average dispersion for all channels is about −2.25 ps/nm, with a channel bandwidth of about 3.1 nm, so as to support typical applications like 400 G coarse wavelength division multiplexer four lanes (CWDM4).

## 2. Principle of CMG-CDC

Figure 1 shows the schematic principle of our proposed CMG-CDC. The devices are designed based on a silicon-on-insulator (SOI) platform with a 220 nm thick top silicon layer so as to support the CMOS-compatible fabrication. The CMG-CDC for multichannel dispersion compensation is realized based on the fundamental single mode chirped-grating-assisted counter-directional coupler (CG-CDC), which is shown in Figure 1a.

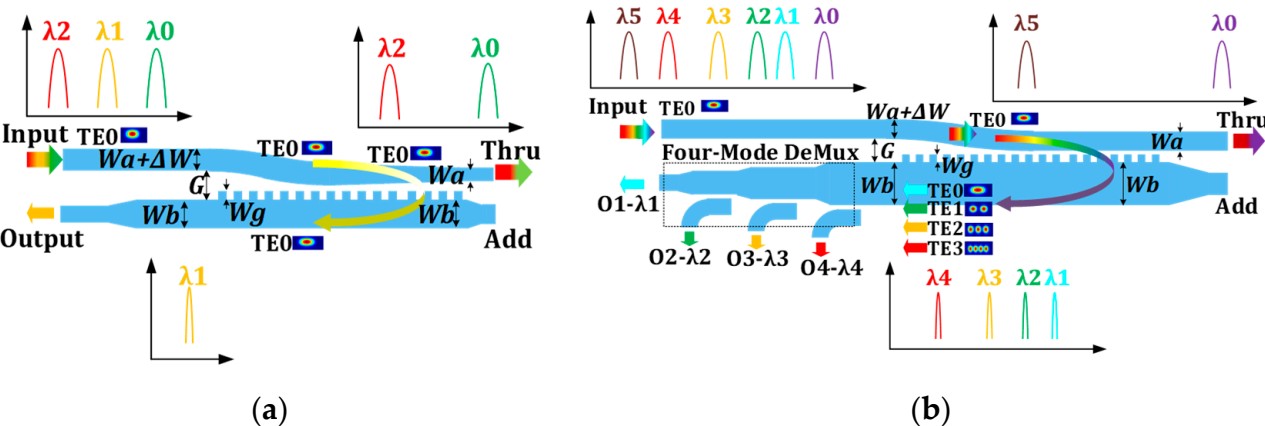

**Figure 1.** Schematics of fundamental single mode CG-CDC (**a**) and four-channel CMG-CDC (**b**).

The fundamental device is mainly composed of two parallel waveguides of different widths, so that asymmetry will be introduced to suppress the codirectional coupling between the waveguides. The grating structure between waveguides provides dielectric perturbations, so the optical input from the upper waveguide will be coupled backward to the below waveguide at wavelengths meeting the phase-matching condition:

$$\left[n_{effa}(z) + n_{effb,i}(z)\right]/2 = \lambda_{c,i}(z)/(2\Lambda) \tag{1}$$

In Equation (1), $n_{effa}$ is the effective index of the forward propagation mode of the upper waveguide, and $n_{effb,i}$ is the backward propagation mode of the below waveguide. $\lambda_{c,i}$ is the resonant wavelength and $\Lambda$ is the period pitch of the grating. If the width of the waveguide is modified, the effective index will be tuned as well. Considering that the width modification of the waveguide is more robust than modifying the waveguide period, we can then apply a linear variation in the upper waveguide from $W_a + \Delta W$ to $W_a$. The below waveguide width is $W_b$.

As the waveguides taper down along with the CMG-CDC, the phase-matching condition shown in Equation (1) shifts to shorter wavelengths. Therefore, shorter wavelengths travel longer distances before being dropped to the output.

Likewise, as shown in Figure 1b, by using a wider below waveguide, which supports two or four modes, and an ADC as a mode demultiplexer, we can expand the dispersion

compensator to two or four channels without cascading two or four devices. The optical input will be coupled into the backward-propagating TE0 and TE1 modes in the below multimode waveguide at different wavelengths. Considering the different effective indices of TE0 and TE1 mode in the below multimode waveguide, the different wavelength signals will then be coupled into O1 and O2 ports, respectively, by a mode demultiplexer, so that wavelength division demultiplexing is simultaneously realized. We just taper the width of the upper single mode waveguide from $W_a + \Delta W$ to $W_a$ and keep the below multimode waveguide with a width of $W_b$. This is because the bandwidth of two channels will be different if we taper the multimode waveguide because the effective indices of the two modes are not sensitive to the waveguide width equally.

The two-mode demultiplexer is based on an ADC structure, which has a narrow silicon waveguide and a wide tapered waveguide in parallel for optical coupling between each other. The effective refractive index of the TE0 mode of the narrow waveguide is designed to be equal to that of the TE1 mode of the wide waveguide. So, the TE1 mode of the wide waveguide will be injected into the TE0 mode of the narrow waveguide to ensure that the two-mode CMG-CDC can support a two-channel dispersion compensation and multiplexer.

More modes can support more wavelength channels, and cascaded tapered directional couplers can be used to split the different modes, as shown in Figure 1. However, there is a problem: we cannot arbitrarily change the effective refractive index of the waveguide so as to make the centering wavelength $\lambda_{c,i}$ distributed as we desired. For example, for a 220 nm high SOI waveguide, no matter how we change its width, we cannot make the effective refractive index of the multimode equal to an arithmetic sequence as we needed. Figure 2a shows calculated intensity distributions of the electric fields for the TE0-like to TE3-like modes of the CMG-CDC. The calculation is carried out by commercial software (Lumerical MODE) based on the finite element method (FEM). Figure 2b shows the calculated effective indices of the modes with the phase-match conditions with corresponding wavelengths with respect to the four-channel CMG-CDC. Thus, we need smaller effective indices for the TE0-like and TE2-like modes for realizing an equal-spaced four-channel operation. To achieve this, we modified the waveguide structure by utilizing the different distributions of various modes in a rectangular waveguide. Its schematic structure is shown in Figure 3a. We chose to engrave a groove with a depth of $H_2 = 70$ nm and a width of $A = 200$ nm on a rectangular multimode waveguide. Those structure parameters are feasible for practical fabrication since they are compatible with most standard manufacturing processes. Based on such modification, the four wavelength channels can be equally placed, as schematically shown in Figure 3b.

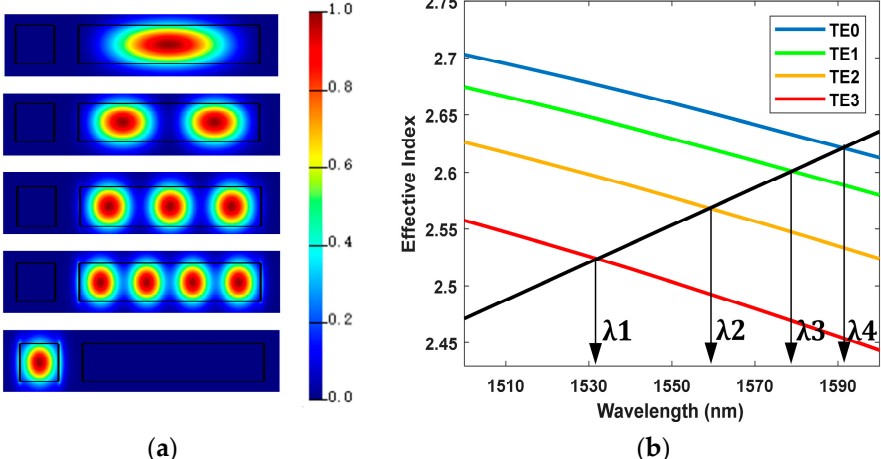

(**a**)            (**b**)

**Figure 2.** (**a**) Intensity distributions of the electric fields for the fundamental TE-like modes of four channel CMG-CDC; (**b**) Effective index of the modes with the phase-match conditions and corresponding wavelengths labeled.

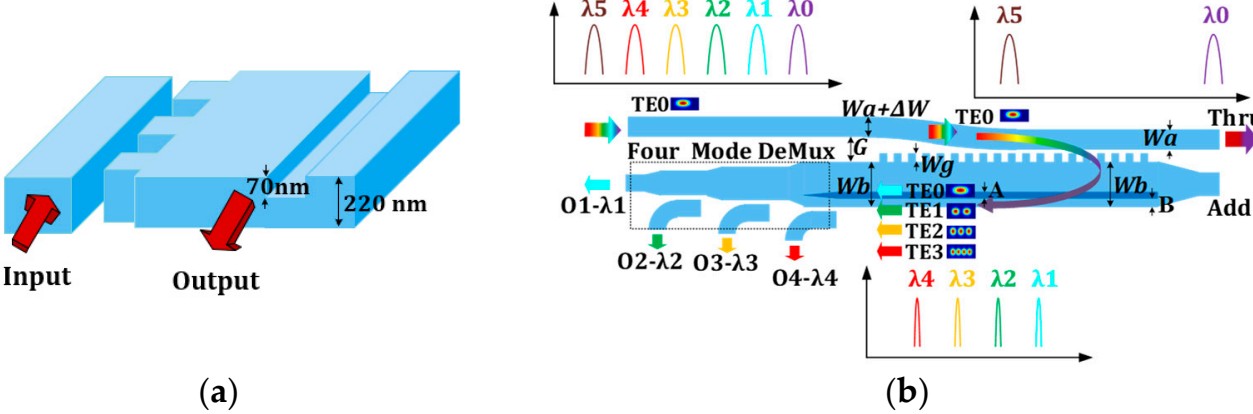

**(a)**                                   **(b)**

**Figure 3.** The schematic structure of the groove-assisted multimode waveguide (**a**) and the operation process with top view of the four-channel CMG-CDC (**b**).

Shown in Figure 4 is the effective indices of TE0 ~TE3 modes, with different groove position B at 1550 nm. To make the $n_{effb,i}$ close to an arithmetic sequence so that the spacing between the center wavelengths of each channel is the approximately equal, B = 610 nm is selected according to the simulation results. As shown in Figure 5a, the electrical intensity of the TE0-like and TE3-like modes in the rectangle waveguide almost remains unaffected since the groove is far enough from their mode fields. At the same time, the TE1-like and TE2-like modes are slightly altered due to the groove structure. By doing so, the effective indices of the TE1-like and TE2-like modes decrease due to the groove, rather than the indices of the TE0-like and TE3-like modes. Thus, we can make the effective index $n_{effb,i}$ close to an arithmetic sequence and make the Bragg wavelength of the four channels of grooved CMG-CDC match the desired values. Figure 4 shows the calculated effective indices of the modes and the Bragg wavelengths of the four-channel CMG-CDC with groove.

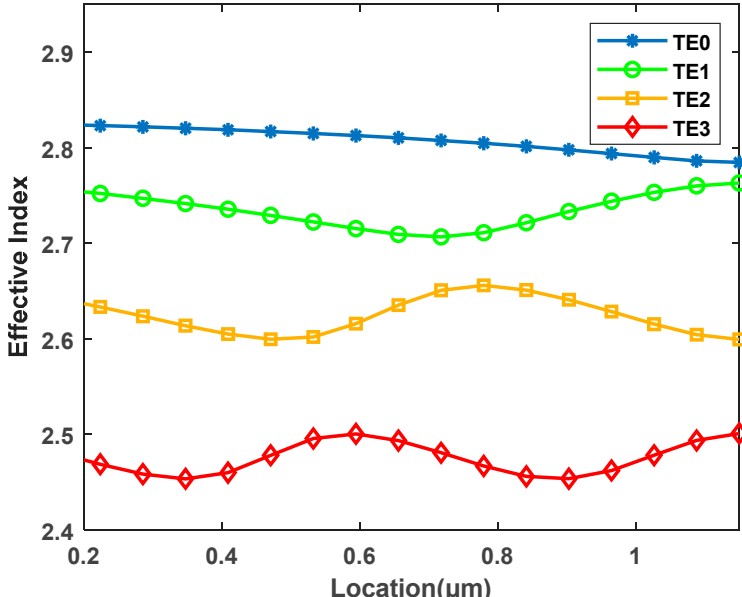

**Figure 4.** Simulated effective indices of the TE0~TE3 modes in multimode waveguides of groove assisted four-channel CMG-CDC with respect to the grooved position B.

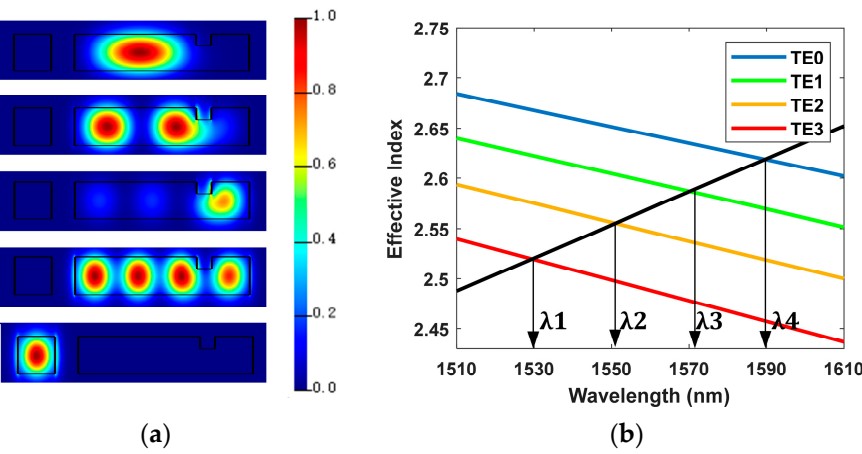

**Figure 5.** (**a**) Simulated intensity distributions of the electric fields for the fundamental TE-like modes of groove assisted four channel CMG-CDC; (**b**) Calculated effective index of the modes and Bragg wavelengths.

## 3. Results and Discussions

During the design of the proposed device, suppression of forward coupling is needed for CG-CDC and CMG-CDC, which is different from the conventional directional coupler, which is a forward coupling-based device. Otherwise, there will be crosstalk between different channels. Therefore, the effective refractive index of the TE0 mode in single mode waveguide should not be too close to that of the modes in the multimode waveguide. In this case, we design the waveguide parameters as shown in Table 1. The period number is 2000 and the length of coupler (*L*) is at most 0.62 mm.

**Table 1.** Geometric parameters of the device.

| Device | Channel Number | $W_a$ (nm) | $\Delta W$ (nm) | $W_b$ (nm) | $B$ (nm) | $A$ (nm) | $H_2$ (nm) | $\Lambda$ (nm) | $W_g$ (nm) |
|---|---|---|---|---|---|---|---|---|---|
| CD-CDC | 1 | 400 | 20 | 620 | NA | NA | NA | 310 | 150 |
| CMG-CDC | 2 | 480 | 20 | 1250 | NA | NA | NA | 256 | 150 |
| Groove CMG-CDC | 4 | 500 | 20 | 2400 | 610 | 200 | 70 nm | 303 | 200 |

In order to suppress the spectral ripple and sidelobes caused by the limited grating length, it is necessary to apodize the coupling strength. Balancing the bandwidth and ripple suppression of the grating, we choose asymmetric apodization, for which the apodization function is only applied on the gratings near the input port, and the ratio of the apodization length is 1/2 of the total CMG-CDC length. Considering the limitation of manufacturing accuracy, we chose to keep the width of the grating perturbation ($W_g$) unchanged and changed the coupling strength by changing the gap spacing between the two waveguides so as to realize the apodization. The function of waveguide spacing *G* with respect to position z is described as follows:

$$G(z) = \begin{cases} G_{min} + H[1 - exp(-a\frac{(z-L/2)^2}{L^2})], z < L/2 \\ G_{min}, z \geq L/2 \end{cases} \tag{2}$$

in which $G_{min}$ is the minimum gap and is chosen to be 200 nm and *H* is chosen to be 2000 nm, which determines the max gap. Coefficient *a* related the apodization strength is designed to be 2.

Figures 6–8 show the reflection and group delay spectra of the proposed CG-CDC and CMG-CDC. This calculation is carried out by commercial software (Lumerical FDTD) based on the finite difference time domain method (FDTD). As shown in Figure 6a, near 1550 nm,

the input signal of TE0 mode in the upper waveguide of CG-CDC is transformed into the TE0 mode of the below waveguide with a loss less than 1 dB and a passband at about 10 nm. Figure 6b shows that, within the passband, the group delay increases as the wavelength decreases with little ripples, and the dispersion in terms of the slope of group delay is about $-2$ ps/nm. At the through port, the transmission spectrum acts as a band-stop profile in the working waveband corresponding to the design, which is complementary compared with that of the reflection port that acts as a band-pass profile.

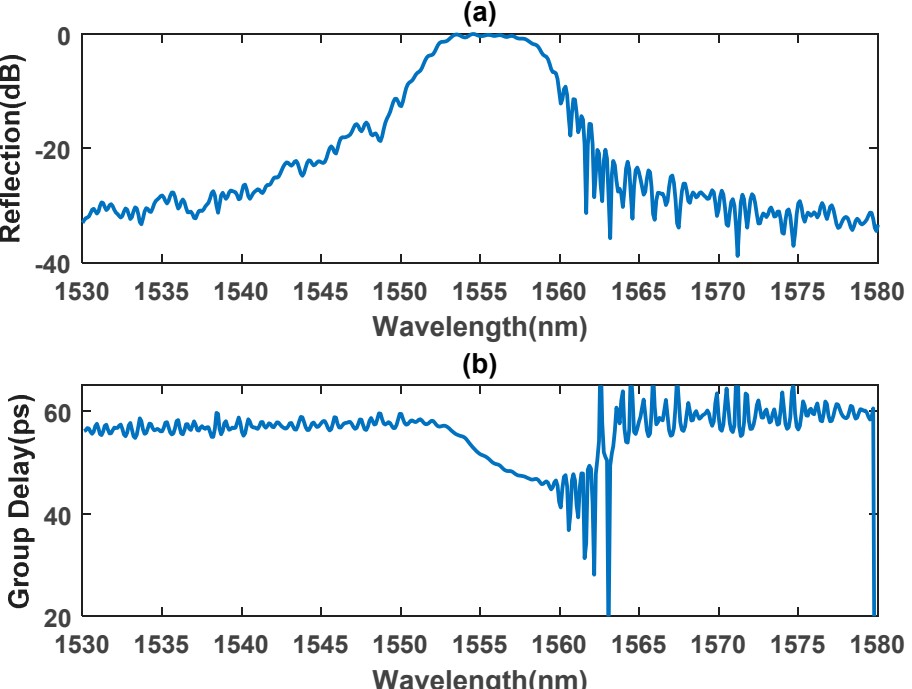

**Figure 6.** Reflection (**a**) and group delay (**b**) for CG-CDC.

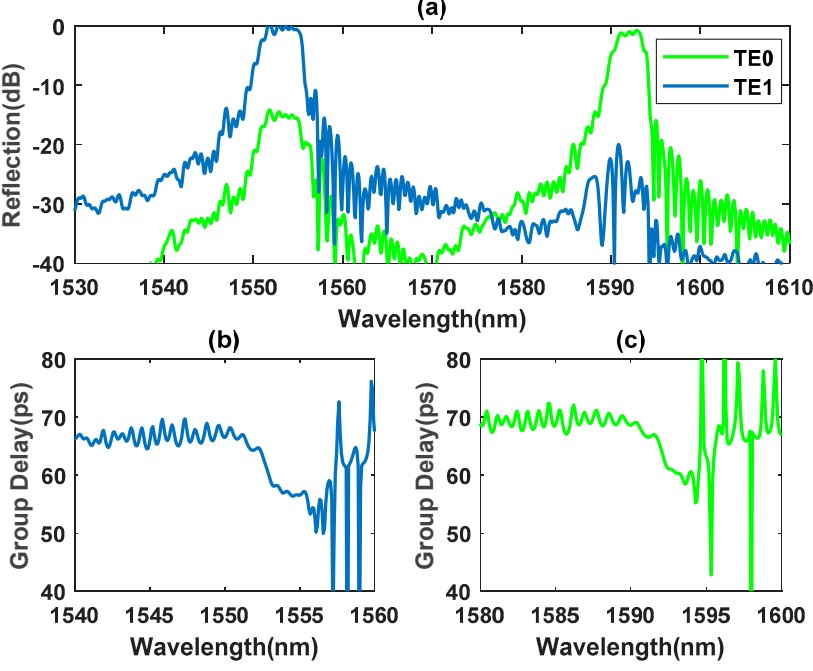

**Figure 7.** Reflection (**a**) and group delay (**b**,**c**) for two-channel CMG-CDC.

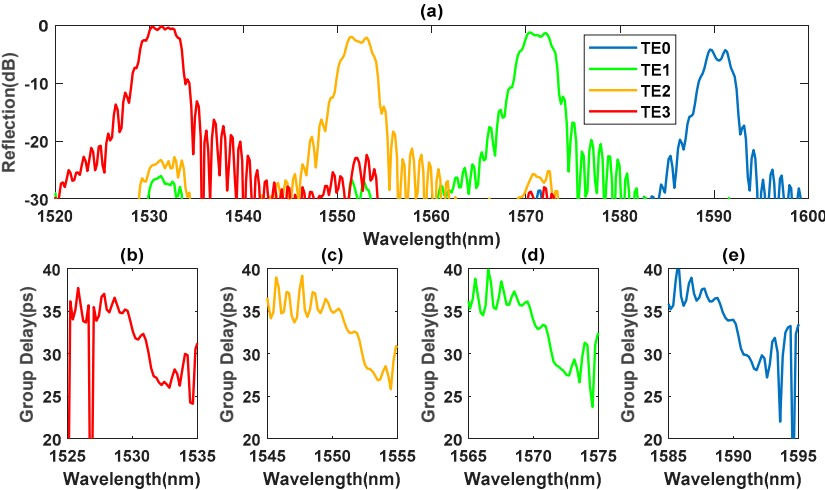

**Figure 8.** Reflection (**a**) and group delay (**b**–**e**) for groove assisted four-channel CMG-CDC.

Figure 7a demonstrates that the two-mode CMG-CDC which can support two-channel dispersion compensation and wavelength multiplexing at wavelengths of 1553 nm with 1.2 dB loss and 1593 nm with 2.1 dB loss. The bandwidth is about 5 nm at 1553 nm and 4 nm at 1593 nm, and the dispersion is −3.1 ps/nm and −3.8 ps/nm, as shown in Figure 7b. As for the four-channel situation shown in Figure 8a, four-mode grooved CMG-CDC can drop the input signal to the TE0~TE3-like modes of multimode waveguide at 1530 nm, 1550 nm, 1570 nm, and 1590 nm, with passbands of 4 nm, 2.8 nm, 3.2 nm, and 2.4 nm and losses of 4.2 dB, 1.2 dB, 2.0 dB, and 0.1 dB. Crosstalk is less than −20 dB, mainly caused by the codirectional coupling of the two waveguides. Figure 8b–e show the group delay of the four channels with dispersion calculated to be −1.6 ps/nm, −2.5 ps/nm, −2.1 ps/nm, and −2.8 ps/nm, respectively. The average dispersion is −2.25 ps/nm, which is still small, but it can be easily increased by enlarging the total length of the device.

## 4. Conclusions

In this work, we demonstrated a multi-channel dispersion compensation with wavelength division MUX/DeMUX based on an on-chip device named CMG-CDC. A device design for four-channel CMG-CDC at the C/L band is presented, with a channel spacing of 20 nm assisted by a grooved multimode waveguide structure. The average dispersion for all channels is about −2.25 ps/nm with a channel bandwidth of about 3.1 nm. The device is highly compact and highly scalable, which makes it rather convenient for increasing the group velocity dispersion (GVD) and channel number, indicating flexible applications for versatile systems, including typically CWDM4 transceivers.

**Author Contributions:** Conceptualization, Z.L., J.D. and Z.H.; methodology, Z.L.; investigation, Z.L.; writing, Z.L., J.D. and Z.H.; supervision, J.D. All authors have read and agreed to the published version of the manuscript.

**Funding:** This research was funded by National Key R&D Program of China (2023YFB2905502), National Natural Science Foundation of China (62122047, 61935011).

**Institutional Review Board Statement:** Not applicable.

**Informed Consent Statement:** Not applicable.

**Data Availability Statement:** Data are contained within the article.

**Conflicts of Interest:** The authors declare no conflicts of interest.

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
