# Peer review of "On-Chip Multichannel Dispersion Compensation and Wavelength Division MUX/DeMUX Using Chirped-Multimode-Grating-Assisted Counter-Directional Coupler"

_photonics, doi:10.3390/photonics11020110_

Round 1

Reviewer 1 Report

Comments and Suggestions for Authors

In this work, the authors present a novel on-chip device design with two useful functions together, realizing multichannel wavelength MUX and DEMUX, as well as dispersion compensation. The whole device is compact and scalable. The design for one, two, and four channel WDM operation is carried out and theoretically investigated with low loss and low crosstalk. The groove assisted structure is utilized for equally spacing the operation channels based on mode field manipulation, which is quite a good idea. This work is generally novel and highly suitable for publication on Photonics. Here bellow are several minor revision comments for the authors to better improve the paper.

1, Some figures’ quality can be improved, such as the electric field distribution.

2, Can the authors explain why design this device at C and L band? Is it because the dispersion compensation demand is higher here since SMF has higher dispersion? But to my understanding, CWDM4 for optical interconnection is mainly at O band, as the very high speed up to single lane 112Gbps also need certain dispersion compensation for long distance applications like FR, LR, and even ER. Please briefly discuss.

3, Can this design work on different platforms?

Reviewer 2 Report

Comments and Suggestions for Authors

Dear Authors,

Your work is interesting to me.  I want to offer the following suggestions.

1. Consider changing the "contradictional" term to "counter-directional".  This terminology is more appropriate for couplers in both optics and RF.

2. Include the wavelength range for the "C/L band" in the Abstract and Introduction for non-communications readers.

3. GVD and CWDM4 are not defined when used in the Abstract and Introduction.

4. Figure 1 is difficult to read and needs to be enlarged.  Texts are too small to read.  The contrast of black text on blue is no good.

5. Variables in text, e.g. n, l, and w, should have the same format as in the equation.

6. Texts on Figure 2(b) are difficult to read.

7. Figure 3 has problems similar to those in Figure 1.

8. Figure 4 has problems similar to those in Figure 2(b).

9. Figure 5 has no (a) and (b).  Texts on the graph are challenging to read.

10. The format of Equation 2 is different to that of Equation 1.  Please confirm and be consistent.

11. "wavelength" in Figures 6, 7 and 8 should be "Wavelength".  The numbers on the axes should be bold to make it easier to read.

12. Relating to the comment in the last sentence before the Conclusion that -2.25 ps/nm is still small to be useful.  What are useful in terms of negative dispersion required, and how big must the device be to achieve it?

Comments on the Quality of English Language

Please see the comments to the authors.

Round 2

Reviewer 2 Report

Comments and Suggestions for Authors

Dear Authors,

Thank you for addressing all reviewers' comments.

Best regards.

Comments on the Quality of English Language

It's worth a last spelling and grammar check before print.